# Integrating all-optical switching with spintronics

M.L.M. Lalieu[1], R. Lavrijsen[1] & B. Koopmans [1]

All-optical switching (AOS) of magnetic materials describes the reversal of the magnetization using short (femtosecond) laser pulses, and received extensive attention in the past decade due to its high potential for fast and energy-efficient data writing in future spintronic memory applications. Unfortunately, the AOS mechanism in the ferromagnetic multilayers commonly used in spintronics needs multiple pulses for the magnetization reversal, losing its speed and energy efficiency. Here, we experimentally demonstrate on-the-fly single-pulse AOS in combination with spin Hall effect (SHE) driven motion of magnetic domains in Pt/Co/Gd synthetic-ferrimagnetic racetracks. Moreover, using field-driven-SHE-assisted domain wall (DW) motion measurements, both the SHE efficiency in the racetrack is determined and the chirality of the optically written DW's is verified. Our experiments demonstrate that Pt/Co/ Gd racetracks facilitate both single-pulse AOS as well as efficient SHE-induced domain wall motion, which might ultimately pave the way towards integrated photonic memory devices.

---

[1] Department of Applied Physics, Institute for Photonic Integration, Eindhoven University of Technology, P.O. Box 5135600 MB Eindhoven, The Netherlands. Correspondence and requests for materials should be addressed to M.L.M.L. (email: m.l.m.lalieu@tue.nl)

The high potential of all-optical switching (AOS) for fast and energy-efficient memory devices was quickly recognized after it was first discovered in GdFeCo alloys about a decade ago[1]. The AOS allows to directly store optical information in magnetic bits without the need of energy-costly electronics, showing high potential to be used in future photonic integrated circuits. Moreover, the switching speed for the AOS is on the picosecond time scale, much faster than conventional switching mechanisms that operate in the (sub) nanosecond regime. Furthermore, the AOS is very energy efficient, needing only tens of fJ to switch a $50 \times 50$ nm$^2$ sized bit[2]. With the spintronic integration in prospect, the discovery of AOS initiated a rapidly developing field of research, initially aimed at unravelling the mechanism of this ultrafast switch. Soon it was discovered that the AOS in rare earth-transition metal (RE-TM) alloys is a purely thermal single-pulse process[3,4], and that an earlier observed helicity dependence was the result of magnetic circular dichroism[1,5]. The research field gained an additional boost when AOS was observed in ferromagnetic thin films and multilayers[6], which are material systems already heavily used in the field of spintronics for future memory devices such as the racetrack memory[7–9] and next-generation magnetic random access memory[10]. Unfortunately, the helicity-dependent AOS found in these materials turned out to be a cumulative process needing multiple pulses[11,12], preventing its use in fast spintronic devices. Clearly, the thermal single-pulse AOS mechanism is needed for successful spintronic integration. Although this mechanism is well established in RE-TM alloys, spintronic devices like the racetrack memory rely on interface-induced phenomena inherent to multilayered ultra-thin-film structures. In our recent work, we have shown efficient thermal single-pulse AOS in such multilayers, made of a Pt/Co/Gd synthetic-ferrimagnetic stack[2]. This structure was chosen because of; (i) the interfacial anti-ferromagnetic coupling between the Co and Gd layers[13], (ii) the large contrast in demagnetization times between the Co and Gd[14,15], (iii) the Pt seed layer induced perpendicular magnetic anisotropy, and (iiii) the built-in interfacial Dzyaloshinskii-Moriya interaction (iDMI)[13].

Here, we experimentally demonstrate that the Pt/Co/Gd stack is indeed an ideal candidate to facilitate the integration of AOS with spintronics, and more specifically with the racetrack memory. We do this by demonstrating clear and robust single-pulse AOS in Pt/Co/Gd racetracks, and verifying that the DW's of the optically written domain are chiral Néel walls that can be moved coherently along the racetrack by means of the SHE. Moreover, the SHE efficiency in the Pt/Co/Gd racetrack is determined, predicting high DW velocities, and a proof-of-concept measurement is presented demonstrating on-the-fly data writing in the racetrack, i.e. showing single-pulse AOS while simultaneously sending an electrical current through the racetrack that transports the written magnetic domains by means of the SHE.

## Results

### Sample structure and characterization
The measurements were performed on Ta(4)/Pt(4)/Co(1)/Gd(3)/Pt(2) stacks (thickness in nm), which were deposited on a Si/SiO$_2$(100 nm) substrate using DC magnetron sputtering (see Methods). The samples were patterned into 5 μm wide and 90 μm long wires using electron beam lithography and argon ion milling. Each wire contains typically two sets of lateral legs (2 μm wide) forming a Hall cross, used to measure the out-of-plane magnetization by means of the anomalous Hall effect (AHE). All structures showed perpendicular magnetic anisotropy with square out-of-plane hysteresis loops and 100% remanence.

### Deterministic single-pulse AOS in Pt/Co/Gd racetracks
First, the AOS in the magnetic wires was investigated by measuring the magnetization in one of the Hall crosses while at the same time it was exposed to a train of linearly polarized laser pulses ($\approx$100 fs). The magnetization in the cross was measured using the AHE[11,16] (see Methods), as illustrated in Fig. 1a. The AHE signal is proportional to the out-of-plane component of the magnetization in the Hall cross area, which was normalized to the up (+1) and down (−1) saturation values using an external out-of-plane magnetic field. A typical measurement (without any external field) is shown in Fig. 1b. In order to clearly identify the single pulses in the AHE measurement a relatively low laser-pulse repetition rate of 0.5 Hz was used. It is clearly seen that the magnetization in the Hall cross region toggles between the saturated up (+1) and down (−1) states at the frequency of the incoming laser pulses. Repeating the measurement for a longer time demonstrated a 100% success rate of the AOS for over more than 5000 subsequent laser pulses. This shows that indeed a deterministic single-pulse all-optical switch of the magnetization is present in the patterned Pt/Co/Gd wires.

The result presented in Fig. 1b shows full AOS in the Hall cross. The laser spot was larger than the Hall cross, meaning that the area of the Pt/Co/Gd wire that was exposed to the laser pulse was larger than the region probed by the Hall cross. It is known that depending on the laser fluence, a multidomain state can form at the centre of the (Gaussian shaped) laser spot, in which case only AOS is observed in an outer rim of the excited area[11]. In view of the transport measurements discussed in the following, it is important that a single homogeneous domain is written in the wire by the laser pulse. Supplementary Note 1 shows measurements on the AOS as a function of the overlap of the laser spot and the Hall cross, demonstrating that indeed homogeneous domains were written in the Pt/Co/Gd wire.

### On-the-fly all-optical data writing
With the AOS in the wires verified, transport measurements on the optically written domains were performed. The DW's in the Pt/Co/Gd wire are expected to be chiral Néel walls due to the iDMI[13]. Such chiral Néel walls can be moved coherently through the wire using an electrical current, exploiting the SHE in the heavy-metal Pt seed layer of the Pt/Co/Gd wire. The SHE in the Pt layer results in a spin accumulation at the Pt/Co interface that generates a torque on the DW that causes it to move. The direction of the DW motion is determined by the sign of the SHE and the chirality of the DW. For a ferromagnet on top of a Pt layer, the SHE driven DW motion is reported to be along the current direction, i.e. against the electron flow direction[17,18].

Combining the SHE-driven transport of the optically written domains with the single-pulse AOS in the racetrack, we have been able to establish on-the-fly data writing. In such measurement, illustrated in Fig. 2a, AOS is used to write a domain in a Pt/Co/Gd wire containing two Hall crosses, while at the same time a chosen DC current is flowing through the wire (direction indicated in the figure). Since both DW's enclosing the written domain have the same chirality (shown in Fig. 3c), they will move coherently along the current direction by the SHE as soon as they are written. This means that the full domain will be transported along the wire, passing the Hall cross at the end where it will be recorded using the AHE.

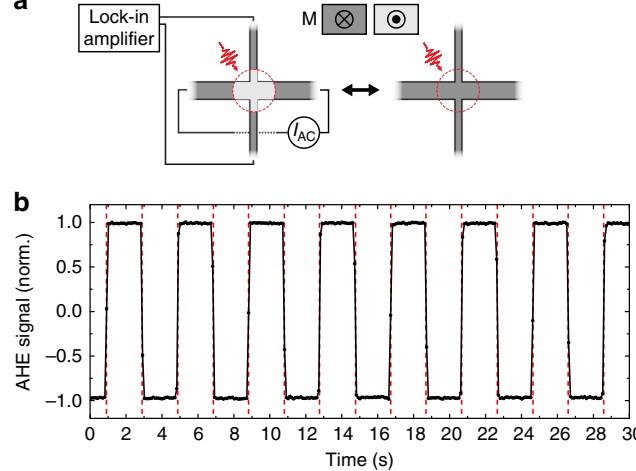

**Fig. 1** Deterministic single-pulse AOS in a Pt/Co/Gd racetrack. **a** Schematic overview of the AOS measurement in a Hall cross on the Pt/Co/Gd wire. A small AC current is applied along the wire, while the resulting anomalous Hall voltage is measured across the legs using a lock-in amplifier. Exciting the cross by subsequent single linearly polarized laser pulses toggles the magnetization in the exposed region (dotted circle) up and down. **b** Measurement of the normalized AHE signal as a function of time during laser-pulse excitation at a repetition rate of 0.5 Hz. No external field was applied during the measurement

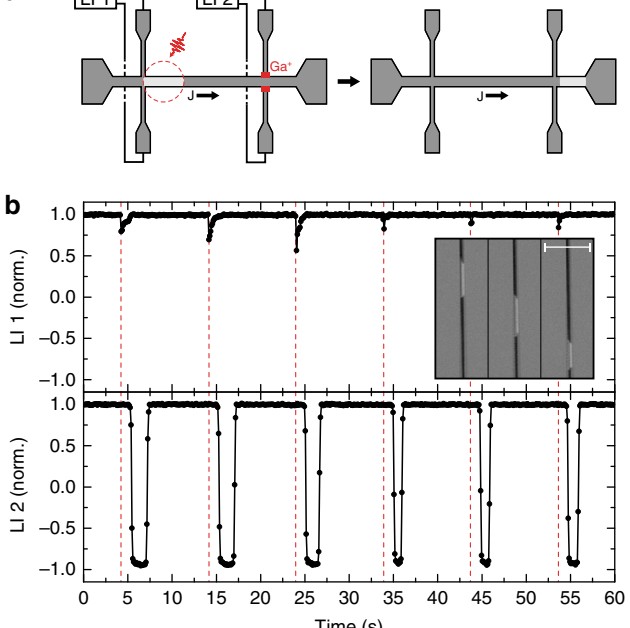

**Fig. 2** On-the-fly AOS with purely SHE-driven DW motion. **a** Illustration of the field-free proof-of-concept measurement, demonstrating on-the-fly single-pulse AOS in combination with purely SHE-driven transport of the optically written domains in a Pt/Co/Gd wire. The AHE signal in the Hall crosses is measured using lock-in amplifiers LI 1 and LI 2. The red dotted circle illustrates the region exposed by the laser pulse, and the red blocks indicate the regions exposed to $Ga^+$ ion irradiation. **b** Results of the measurement illustrated in (**a**), showing the normalized AHE signal of the two Hall crosses as a function of time. The red dotted lines mark the time at which the laser pulse hits the wire and writes the magnetic domain. The inset shows three snapshots (Kerr microscope images) of a magnetic up (white) domain in an otherwise down (black) magnetized Pt/Co/Gd racetrack (2 μm wide), while it was moved through the wire by a train of current pulses. The scale bar in the figure corresponds to 20 μm

As indicated in Fig. 2a (red squares), the legs of the second (right) Hall cross in the Pt/Co/Gd wire were exposed to $Ga^+$ ion irradiation[19]. This was done to magnetically 'cut-off' the legs in order to prevent DW pinning at the entrance of the cross, as is discussed in Supplementary Note 2. To avoid pinning of one of the DW's at the first Hall cross, the laser was aligned not at the centre but at the right side of the first Hall cross. In order to verify if and when a domain is written, a small overlap between laser pulse and the first Hall cross was maintained. Similar measurements performed with the laser aligned completely in between the two Hall crosses and with the laser centred at the centre of the first Hall cross can be found in Supplementary Note 3, of which the latter clearly demonstrates the pinning of the DW at the entrance of the non-irradiated Hall cross.

The result of the measurement with a laser-pulse repetition rate of 0.1 Hz and a DC current of +5.5 mA is shown in Fig. 2b. In the top half of the figure, the normalized AHE signal of the first Hall cross is shown as a function of time. It can be seen that small peaks appear in the signal at the frequency of the laser pulses. These peaks start with a sudden step, corresponding to the (small) exposed part of the Hall cross being switched by the laser, whereafter the signal quickly returns to the saturation value, showing the DW moving out of the cross. More interesting is the AHE signal of the second Hall cross, shown in the bottom half of the figure. It can be seen that shortly after the domain is written (red dotted lines), the magnetization in the second Hall cross switches down (−1) and shortly thereafter switches back up (+1) again. These switches mark the passing DW's, i.e. the passing domain. The time between the two switches, which is a measure of the width of the domain, varies, which is attributed to random pinning of the DW's in the wire. This proof-of-concept measurement demonstrates on-the-fly single-pulse AOS and simultaneous SHE-driven motion of magnetic domains in a single racetrack.

Additionally, the SHE induced DW motion was verified using a Kerr microscope, shown in the inset of Fig. 2b. The figure shows three snapshots of a magnetic up (white) domain

in an otherwise down (black) magnetized Pt/Co/Gd wire (2 μm wide), while it was moved through the wire by a train of current pulses. As can be seen, the DW's indeed move coherently through the wire (against the electron flow), while a change in the domain width is observed as was discussed in the previous measurement.

**Measurement of the SHE efficiency and DW chirality**. A quantitative analysis of the SHE efficiency and DW chirality in the optically written domains was obtained by performing field-driven-SHE-assisted DW velocity measurements, as illustrated in Fig. 3a. In such measurement, the DW motion is driven by an out-of-plane applied field while at the same time a DC current is sent through the wire. Depending on the current polarity, the DW motion is either assisted or hindered via the SHE, resulting in an increase or decrease of the DW velocity, respectively. The Pt/Co/Gd wires used for these measurements contain two Hall crosses that are separated by 60 μm, and the legs of both crosses are exposed to $Ga^+$ ion irradiation (discussed earlier). At the start of the measurement the magnetization in the wire is saturated by the external field, whereafter a (static) field is applied in the opposite direction with an amplitude below the domain nucleation field, but above the DW propagation field. Using a single laser pulse, a domain is written into the wire left of the first Hall cross, see Fig. 3a. The

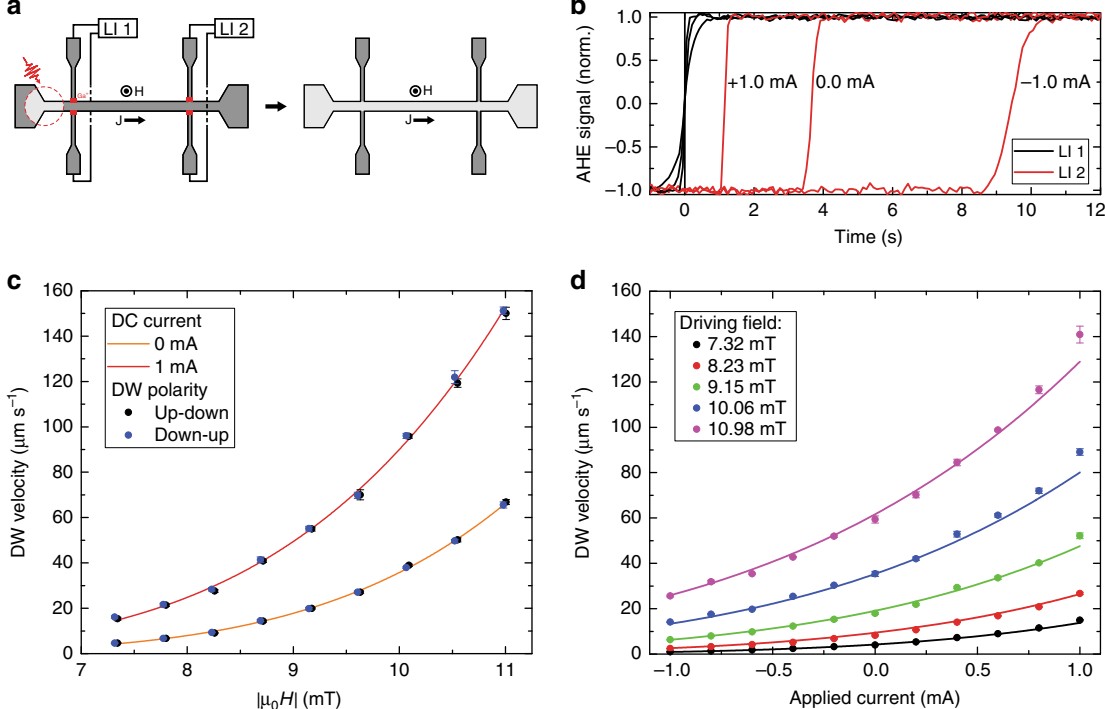

**Fig. 3** Field-driven-SHE-assisted DW velocity measurements. **a** Schematic overview of the field-driven-SHE-assisted DW motion measurement on the Pt/Co/Gd wire. The AHE signal in the Hall crosses is measured using lock-in amplifiers LI 1 and LI 2. The red dotted circle illustrates the region exposed by the laser pulse, and the red blocks indicate the regions exposed to Ga+ ion irradiation. **b** Typical time-of-flight measurements for three different applied current amplitudes, showing the normalized AHE signal in both Hall crosses as a function of time. The step in the signal from −1 (down) to +1 (up) indicates that the DW is passing the specific Hall cross. **c** DW velocity for both DW polarities as a function of the driving field amplitude with 0 and +1 mA of DC current sent through the wire [direction indicated in (**a**)]. The solid lines represent fits to the data following the creep law [Eq. (1)]. **d** DW velocity as a function of the current sent through the wire for different driving fields. Solid lines represent fits to the data using Eq. (1). Error bars in (**c**, **d**) correspond to the standard deviation of the mean using 5 measurements for each data point

applied field will cause the domain to expand through the wire, causing it to pass the two Hall crosses, which will be recorded by a switch in the AHE signal. Using the time of flight and the distance between the two crosses the DW velocity is determined. A typical measurement of the AHE signal in both Hall crosses as a function of time for three different DC current values is shown in Fig. 3b, where the passing of a DW through the Hall crosses is clearly seen in the AHE signal, and the effect of the SHE on the time of flight is apparent.

The DW chirality of the optically written domains was investigated by measuring the effect of the SHE on the DW velocity for both up-down and down-up DW polarities, which were obtained by reversing the saturation and propagation fields. Figure 3c shows the DW velocity for both DW polarities as a function of the driving field amplitude with 0 and +1 mA of DC current sent through the wire (direction indicated in Fig. 3a). The solid lines represent a fit using the creep law for DW motion, which will be discussed later. More important are the observations that the DW velocity is independent of the DW polarity for both current amplitudes, and that a current of +1 mA increases the DW velocity with respect to the case without current. The latter affirms that the current induced contribution to the DW motion is against the direction of electron flow, as was also shown in Fig. 2b, indicating SHE driven DW motion dominated by the bottom Pt/Co interface as discussed earlier. The fact that both DW polarities are moved in the same direction at the same velocity by the SHE confirms that they are chiral Néel walls as discussed earlier, indicating the presence of the iDMI in these wires.

Similar DW velocity measurements, now as a function of the current sent through the wire and for different driving fields $H_{ext}$, were used to determine the SHE efficiency. The results of these measurements are presented in Fig. 3d. The DW motion for the used driving fields of a few millitesla is in the creep regime. In this regime the DW velocity can be described by[20]

$$v = v_0 \exp\left[-\chi\left(\mu_0 H_{ext} \pm \epsilon_{SHE} J_{DC}\right)^{-1/4}\right],\qquad(1)$$

in which $\epsilon_{SHE}$ represents the SHE efficiency, defined as the current density to effective (out-of-plane) field conversion factor, $J_{DC}$ the current density, $v_0$ the characteristic velocity and $\chi$ a scaling factor including the pinning potential and thermal energy. By doing field-driven DW motion measurements, the used current densities could be kept low enough to prevent a significant change in temperature by Joule heating, which was verified by four-point resistance measurements (not shown). As a result, the values of $v_0$ and $\chi$ can be assumed to be independent of the current density, and should be constant in the measurements presented in Fig. 3d. Therefore, the DW velocity as a function of the current density is fitted using Eq. (1), where $H_{ext}$ is known for each curve, and $v_0$, $\chi$ and $\epsilon_{SHE}$ are used as global fit parameters. The fitted value for the SHE efficiency is equal to $\epsilon_{SHE} = 9.73 \pm 0.08$ mT $(10^{11}$ A m$^{-2})^{-1}$ (using a homogeneous current distribution throughout the stack in the current density calculation). This value agrees well with the values found in literature for similar structures using different measurement methods[17,21,22].

Using the measured SHE efficiency a simple prediction can be made for the DW velocity that can be reached when driven by intense nanosecond (ns) current pulses as used in ref. [9]. In their work, the authors used current pulses with a current density up to $30 \times 10^{11}\,\mathrm{A\,m^{-2}}$ and a pulse duration of 5 ns in similar sized wires. With the SHE efficiency of $\approx 9.7\,\mathrm{mT}\,(10^{11}\,\mathrm{A\,m^{-2}})^{-1}$ in the present Pt/Co/Gd wires, this would result in an effective out-of-plane field of $\approx 290\,\mathrm{mT}$. This field can be related to a DW velocity using the work presented in ref. [13], where the field-driven DW velocity is measured in an identical Pt/Co/Gd stack using out-of-plane field pulses with a duration of 20 ns and strengths up to 300 mT. Based on that work, a DW velocity as high as $700\,\mathrm{m\,s^{-1}}$ can be extrapolated for the earlier mentioned current pulses. It is also noted that the DW velocity in the measurement presented in the inset of Fig. 2b was $\approx 7\,\mathrm{m\,s^{-1}}$ (see Supplementary Note 4), which was achieved using current pulses with a current density an order of magnitude lower than used to achieve previously demonstrated DW velocities up to $\approx 1000\,\mathrm{m\,s^{-1}}$[19,23], and fits within the earlier extrapolation. Moreover, it was recently shown that this DW velocity is expected to increase significantly when using a system with a more compensated magnetization[9] or angular momentum[23,24], which can be conveniently optimized in the Pt/Co/Gd racetrack due to the easy engineering of the synthetic-ferrimagnet, e.g., by reducing the Co thickness[2].

In conclusion, we have experimentally demonstrated that both thermal single-pulse AOS as well as SHE induced domain wall motion can be combined in a Pt/Co/Gd racetrack with perpendicular magnetic anisotropy, exploiting the chiral Néel structure of the DW's for coherent and efficient motion of the optically written domains. These results show that the Pt/Co/Gd racetrack is an ideal candidate to facilitate the integration of AOS with spintronics. Moreover, due to the thermal nature of the AOS, the final downsizing of the AOS towards the nanometre scale can be done using plasmonic antenna's, a technique already used in heat assisted magnetic recording (HAMR) to very locally heat the recording material by reducing the FWHM laser-pulse size to $< 40\,\mathrm{nm}$[25,26]. Therefore, the demonstrated proof-of-concept for the integration of AOS with the racetrack memory might pave the way towards integrated photonic memory devices.

## Methods

**Sample fabrication**. The samples used in this work were deposited on Si substrates coated with 100 nm of SiO₂. The deposition was done using DC magnetron sputtering at room temperature. The base pressure in the deposition chamber was $10^{-9}$ mbar. After deposition, Ti/Au contacts were deposited on top of the full sheet sample by a lift-off procedure using UV lithography (ma-N 415 photoresist). Lastly, 5 μm wide and 90 μm long wires were created using electron beam lithography to create a hard mask (ma-N 2410 photoresist) in combination with argon ion milling.

**Measurement techniques**. The AOS was achieved using linearly polarized laser pulses with a central wavelength of 700 nm and a pulse duration of ≈100 fs. The pulse energy used throughout the work presented in the Letter was ≈12 nJ, while the laser spot had a radius (1/e Gaussian pulse) of ≈20 μm.

The out-of-plane magnetization in the Hall crosses was measured using the AHE. To measure the AHE signal a small AC current (150 μA) is sent through the wire, and the resulting anomalous Hall voltage across the lateral legs is measured using a lock-in amplifier.

In case of the Kerr microscope images a differential technique was used. In this technique first an image of the magnetic wire with the magnetization saturated in either the up or down direction was captured. This (background) image was then subtracted from the subsequent images in order to enhance the magnetic contrast.

## Data availability

The data that support the plots within this paper and other findings of this study are available from the corresponding authors upon reasonable request.

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

## Acknowledgements

This work is part of the Gravitation program 'Research Centre for Integrated Nano-photonics', which is financed by the Netherlands Organisation for Scientific Research (NWO).

## Author contributions

M.L.M.L. fabricated the structures, designed and conducted the experiments and ana-lysed the data. B.K. and R.L. supervised the project. M.L.M.L. wrote the manuscript. All authors contributed to the discussion and commented on the manuscript.

## Additional information

**Competing interests:** The authors declare no competing interests.

