## [Peer Review File · Nature Communications]

Reviewers' comments:

Reviewer #1 (Remarks to the Author):

This manuscript reports a novel and creative combination of all-optical magnetic switching with domain wall propagation in a magnetic "racetrack" for possible applications in photonic memory technology. The results are an impressive and convincing demonstration of the basic concept. The field of all-optical switching is clearly moving in the direction of practical applications and this paper will be an important contribution in this regard.

Reviewer #2 (Remarks to the Author):

This manuscript reported an experiment applying all-optical switching (AOS) of ferrimagnetic materials (Pt/Co/Gd) into racetrack memories. They successfully demonstrated that the synthetic-ferrimagnetic stack of Pt/Co/Gd can be flipped by the thermal impact from a single 100fs laser pulse. They also demonstrated the on-the-fly single-pulse AOS with spin Hall effect-driven domain wall motion. As claimed in the manuscript, the domain wall motion speed can be very high in the order of hundreds of m/s which makes racetrack memory a potential candidate for next generation of high speed memory device. They also showed the measurements of the domain wall velocities with the help of external magnetic field and get the SHE efficiency. The parameters extracted from the curve fitting seem consistent with other works. This work has its novelty by combining AOS with racetrack memory, however this work is more appropriate to be published elsewhere rather than Nature Communications. We have some comments below.

1. What's the practical advantage of combining the AOS scheme with the racetrack memory? As explained in ref. 10 "Magnetic domain-wall racetrack memory", recording in the racetrack memory can be done in several mature ways, such as using the self-field of currents passed along neighboring metallic nanowires; using the spin-momentum transfer torque effect derived from current injected into the racetrack from magnetic nanoelements; or using the fringing fields from the controlled motion of a magnetic domain wall in a proximal nanowire writing element.
2. What are the limiting factors for domain wall motion speed for the AOS-compatible materials? The demonstrated speed is several orders of magnitudes slower than others' work. This drastically limits both the recording and reading speeds of the device. The paragraph starting from line 284 only shows some simple prediction and numbers in others' work.
3. The stable domain size (or bit size) in this approach is in the range of 20um and much larger than other writing schemes at the 10s-100s nm range. It is not convincing that the approach can create high-density data recording devices.
4. What is the Curie temperature of the recording material? The AOS in their ferrimagnetic material is a thermal process as the authors previously reported. However, one of the key challenge for racetrack memory device is the large heat generation caused by the high current density required to drive a fast motion of the domain wall. How practical is it to use a large current to achieve high speed domain wall motion using their approach?
5. They need to add pinning structures to improve the reliability of the domain wall motion as others did. As shown in figure 2(b), the motion of domain wall is not as reliable as reported by others.

Reviewer #3 (Remarks to the Author):

Lalieu et al. demonstrate all-optical switching (AOS) in Pt/Co/Gd synthetic ferrimagnetic film and field-driven spin Hall effect assisted domain wall motion of the optically switched domains. It's a neat and interesting demonstration. However, this reviewer is unconvinced about the scientific or technical signification of this work. It appears to be a mere subsequent addition of two unrelated technical demonstrations, AOS and SHE-assisted domain wall motion. Therefore, this reviewer does not support publication in Nature Communications.

Additional comments:

1. The authors' previous demonstration (Ref. 9) of single-pulse AOS in Pt/Co/Gd system is very interesting. The current work utilized the SHE in the Pt layer, which is an interesting approach. However, the domain motion is very slow, unmatched to the speed of AOS. So it is unclear how the driven domain movement is used along with AOS as a means of writing to realize fast magnetic memory or storage application.

2. Fig. 1 shows AOS in Pt/Co/Gd, Fig. 2 shows SHE driven DW motion, Fig. 3 shows field driven DW-motion with measurement of DW velocity. These three measurements are not necessarily connected to each other, they can be done separately. For example, the domains can be generated by applying an external magnetic field without using AOS at all to generate results in Fig. 2 and 3.

3. The authors fail to cite the original works of using AHE to measure AOS effect as listed below:

L. He, J.-Y. Chen, J.-P. Wang, and M. Li, "All-optical switching of magnetoresistive devices using telecom-band femtosecond laser," *Appl. Phys. Lett.* 107, 102402 (2015).

M. S. El Hadri, P. Pirro, C. H. Lambert, N. Bergéard, S. Petit-Watelot, M. Hehn, G. Malinowski, F. Montaigne, Y. Quessab, R. Medapalli, E. E. Fullerton, and S. Mangin, "Electrical characterization of all-optical helicity-dependent switching in ferromagnetic Hall crosses," *Appl. Phys. Lett.* 108, 092405 (2016).

Subject
Rebuttal

Date
15 October 2018

Response to the reviewer's comments:

Reviewer #1 (Remarks to the Author):

This manuscript reports a novel and creative combination of all-optical magnetic switching with domain wall propagation in a magnetic "racetrack" for possible applications in photonic memory technology. The results are an impressive and convincing demonstration of the basic concept. The field of all-optical switching is clearly moving in the direction of practical applications and this paper will be an important contribution in this regard.

We thank **reviewer #1** for his/her analysis of our work, and are pleased to hear that he/she considers our results to be an impressive and convincing demonstration of the basic concept of the combination of all-optical switching with the racetrack memory for possible applications in photonic memory technology. Moreover, we appreciate his/her notion that our paper will be an important contribution to the advancement of all-optical switching towards practical applications.

Reviewer #2 (Remarks to the Author):

This manuscript reported an experiment applying all-optical switching (AOS) of ferrimagnetic materials (Pt/Co/Gd) into racetrack memories. They successfully demonstrated that the synthetic-ferrimagnetic stack of Pt/Co/Gd can be flipped by the thermal impact from a single 100fs laser pulse. They also demonstrated the on-the-fly single-pulse AOS with spin Hall effect-driven domain wall motion. As claimed in the manuscript, the domain wall motion speed can be very high in the order of hundreds of m/s which makes racetrack memory a potential candidate for next generation of high speed memory device. They also showed the measurements of the domain wall velocities with the help of external magnetic field and get the SHE efficiency. The parameters extracted from the curve fitting seem consistent with other works. This work has its novelty by combining AOS with racetrack memory, however this work is more appropriate to be published elsewhere rather than Nature Communications. We have some comments below.

We thank **reviewer #2** for his/her analysis of our work, and we would like to provide a point-by-point response to the reviewer's comments. Changes made to the manuscript are mentioned below, and are highlighted in the revised manuscript.

1. *What's the practical advantage of combining the AOS scheme with the racetrack memory? As explained in ref. 10 "Magnetic domain-wall racetrack memory", recording in the racetrack memory can be done in several mature ways, such as using the self-field of currents passed along neighboring metallic nanowires; using the spin-momentum transfer torque effect derived from current injected into the racetrack from magnetic nanoelements; or using the fringing fields from the controlled motion of a magnetic domain wall in a proximal nanowire writing element.*
- We do not position AOS as a method to improve magnetic racetrack memory. Rather, we propose that the combination of AOS and racetrack provides an entirely new class of hybrid photonic-magnetic devices in which information can be directly coupled from the photonic to the magnetic domain without any intermediate electronic steps.

Seen from this perspective, additional advantage lies both in the speed and energy efficiency of the AOS. The switching speed for the thermal single-pulse AOS is on the picosecond time scale, much faster than conventional switching schemes using fields or spin-transfer-torque, which are on the (sub)nanosecond time scale.

Additionally, the switch is very energy efficient, only needing a few fJ to switch a 20x20 nm² bit [1]. This is much lower than the energy needed for field or spin current induced magnetization reversal [(sub)pJ – nJ] [2]. Moreover, this only concerns the energy required for the actual writing process. In the rapidly developing field of (integrated) photonics, energy costly optical-to-electronic conversions are needed when the optical data needs to be stored/buffered for some time (which is hard to do in optical form). Using the single-pulse AOS scheme presented here, optical data can be directly converted to magnetic bits without the use of any electronics. Read-out can also be done all-optically, e.g., using the well-known and extensively used magneto-optical Kerr effect (MOKE).

As a result of these considerations, we believe that the combination of the thermal single-pulse AOS and the racetrack memory has a big practical advantage with respect to future spintronics devices, and can be considered to be a vital component (optical buffer memory/shift register) in future photonic integration.

[1] M.L.M. Laliu et al., Phys. Rev. B **96**, 220411(R) (2017)

[2] Stupakiewicz et al., Nature **542**, 71-74 (2017)

Change made to manuscript: A more clear discussion on the advantages of the used combination of AOS with the racetrack memory is added to the introductory part of the manuscript.

2. *What are the limiting factors for domain wall motion speed for the AOS-compatible materials? The demonstrated speed is several orders of magnitudes slower than others' work. This drastically limits both the recording and reading speeds of the device. The paragraph starting from line 284 only shows some simple prediction and numbers in others' work.*

- We recognize that the shown domain wall speeds are much lower than seen in other works. As motivated in the manuscript, we use these low velocities to allow for a straight forward and reliable determination of the SHE efficiency and a clear interpretation of the “on-the-fly” concept. As mentioned by reviewer #1, we believe that this enabled us to show a convincing demonstration of the basic concept.

In addition, we do have data showing higher domain wall velocities up to ≈ 7 m/s, already 5 orders of magnitude higher than shown in the manuscript. We have added these data to the supplementary information. The obtained velocity of ≈ 7 m/s is achieved using a current density an order of magnitude lower than used for the high velocities in other works. Furthermore, the obtained velocity also fits within the extrapolation made in the manuscript that predicts the high domain wall velocities, which might be more acceptable coming from this ≈ 7 m/s.

Furthermore, as discussed in the manuscripts, it has been shown in recent works that the domain wall velocity dramatically increases when the ferrimagnet is at the angular momentum compensation temperature. After submission of our manuscript, another paper has been published [3], showing unprecedented high domain wall velocities up to 1300 m/s in micron sized tracks made of Pt/CoGd/TaOx. Our use of synthetic ferrimagnets enables easy material engineering that allows to optimize the angular momentum compensation point (getting it to room temperature), while maintaining the single-pulse AOS capability.

Based on these considerations, we believe that the domain wall velocity is not a limiting factor when it comes to the reading and writing speeds in future data applications.

[3] Caretta et al., Nat. Nanotechnol. (2018) (DOI: 10.1038/s41565-018-0255-3)

Change made to manuscript: Notion of the higher domain wall velocity measurement (≈ 7 m/s) is added to the manuscript (referring to the suppl. information), and the measurement is added to the supplementary information. Also, a citation to the recent publication of high domain wall velocities in the CoGd alloy is added to the manuscript at line 302.

3. *The stable domain size (or bit size) in this approach is in the range of 20um and much larger than other writing schemes at the 10s-100s nm range. It is not convincing that the approach can create high-density data recording devices.*

The size of the optically written domains/bits in the manuscript is not equal to the minimum stable domain/bit size. The minimum size of stable domains is determined by micromagnetic factors. In this respect, the used material is considered to be ideal for high-density data storage due to the low net magnetization of the ferrimagnet. When talking about the size of the optically written domains, we stress that it is determined by both the diameter and the energy of the spatial Gaussian pulse shape of the laser pulse used for the AOS. This can be seen in our previous work [1], where the written domain size strongly decreases with the laser-pulse energy. There is no fundamental (optical) limitation to decrease the bit size towards the 10s-100s nm range needed in the final application.

A valid question we did not discuss in the manuscript is how the final downsizing of the AOS towards the 10s-100s nm scale can be performed. This could be done using diffraction limited laser spots with an energy just a little above the switching threshold. Moreover, one could exploit a technique already well-developed for heat assisted magnetic recording (HAMR) [4]. HAMR uses plasmonic antenna's to very locally heat the material by reducing the FWHM laser beam size to < 40 nm [4]. Since the single-pulse AOS present in the Pt/Co/Gd racetrack is a purely thermal process, the same antenna can be used to write 10s-100s nm sized domains in a nano-sized racetrack.

[1] M.L.M. Laliu et al., Phys. Rev. B **96**, 220411(R) (2017)

[4] B.C. Stipe et al., Nat. Photon. **4**, 484-488 (2010)

Change made to manuscript: A reference to the work on HAMR demonstrating the reduced FWHM spot size of < 40 nm is added to the manuscript, and is related to the final downsizing of the AOS towards the nanometer scale.

4. *What is the Curie temperature of the recording material? The AOS in their ferrimagnetic material is a thermal process as the authors previously reported. However, one of the key challenge for racetrack memory device is the large heat generation caused by the high current density required to drive a fast motion of the domain wall. How practical is it to use a large current to achieve high speed domain wall motion using their approach?*
- We agree that heating plays an important role both in AOS and domain wall motion. However, we believe this is of no limiting factor in our approach and is a matter of final engineering, which goes beyond our demonstration of the basic concept.

To give some reasoning for this viewpoint, we refer to Fig. 3(b) of our previous paper [1]. The inset of this figure shows a strong decrease in the

minimum energy needed for the AOS with decreasing Co thickness, which is the result of a decrease of the Curie temperature of Co with its thickness. This demonstrates that with the synthetic Co/Gd ferrimagnetic we can easily engineer the Curie temperature while maintaining the single-pulse AOS. Moreover, this suggests that when the racetrack is heated by the current, the additional laser-pulse energy required for the AOS goes down, recycling the current-induced heat to lower the switching energy.

If, on the other hand, the reviewer is afraid that the Curie temperature would be reached merely by the high current density even in the absence of a laser, it would be a trivial solution to enhance the Curie temperature (but this is an issue that would apply to racetrack memories in general).

[1] M.L.M. Lalieu et al., Phys. Rev. B **96**, 220411(R) (2017)

5. *They need to add pinning structures to improve the reliability of the domain wall motion as others did. As shown in figure 2(b), the motion of domain wall is not as reliable as reported by others.*

- As already discussed in the response to comment 2, the domain wall velocity in our work is kept low, which means it is within the *creep regime*. In this regime, the domain wall motion is dominated by pinning. In the case of other works where high domain wall velocities are shown, the domain wall motion is less stochastic because it happens in the *flow regime*, in which pinning plays no significant role. In case of the final applications, the domain wall motion will be in the mentioned flow regime, and is shown to be much more deterministic [5].

If the application in mind would require extrinsic pinning structures, e.g. for synchronization purpose, PMA racetracks provide several ways to do so, but this is beyond the scope of our work.

[5] S-H Yang et al., Nat. Nanotechnol. **10**, 221-226 (2015)

Reviewer #3 (Remarks to the Author):

Lalieu et al. demonstrate all-optical switching (AOS) in Pt/Co/Gd synthetic ferrimagnetic film and field-driven spin Hall effect assisted domain wall motion of the optically switched domains. It's a neat and interesting demonstration. However, this reviewer is unconvinced about the scientific or technical signification of this work. It appears to be a mere subsequent addition of two unrelated technical demonstrations, AOS and SHE-assisted domain wall motion. Therefore, this reviewer does not support publication in Nature Communications.

We thank **reviewer #3** for his/her analysis of our work, and we would like to provide a point-by-point response to the reviewer's comments. Changes made to the manuscript are mentioned below, and highlighted in the revised manuscript.

1. *The authors' previous demonstration (Ref. 9) of single-pulse AOS in Pt/Co/Gd system is very interesting. The current work utilized the SHE in the Pt layer, which is an interesting approach. However, the domain motion is very slow, unmatched to the speed of AOS. So it is unclear how the driven domain movement is used along with AOS as a means of writing to realize fast magnetic memory or storage application.*
 - See the response to comment 2 of reviewer #2. Furthermore, note that with a 50 nm bit size, a domain wall velocity of 1000 m/s corresponds to a 20 Gbit/s data rate.
2. *Fig. 1 shows AOS in Pt/Co/Gd, Fig. 2 shows SHE driven DW motion, Fig. 3 shows field driven DW-motion with measurement of DW velocity. These three measurements are not necessarily connected to each other, they can be done separately. For example, the domains can be generated by applying an external magnetic field without using AOS at all to generate results in Fig. 2 and 3.*
 - We agree with the reviewer that the experiments could in principle be done separately, and the domains could be written using different methods. However, the breakthrough of the presented work is that all the measurements are actually possible in one and the same racetrack, demonstrating the combined properties;
 - a perpendicular magnetic anisotropy,
 - large spin-orbit torques,
 - chiral Néel walls resulting from a sizable interfacial Dzyaloshinskii-Moriya interaction,
 - a synthetic ferrimagnetic arrangement due to an antiparallel coupling between the Co and Gd layers,
 - and significant difference in the laser-induced demagnetization time scale for both magnetic entities– which are all vital ingredients for the desired functionality.

A detailed discussion on the advantage of the presented combination of AOS with the racetrack memory is given in the response to comment 1 of reviewer #2.

3. *The authors fail to cite the original works of using AHE to measure AOS effect as listed below:*
 - a. *L. He et al., "All-optical switching of magnetoresistive devices using telecom-band femtosecond laser," Appl. Phys. Lett. **107**, 102402 (2015).*
 - b. *M. S. El Hadri et al., "Electrical characterization of all-optical helicity-dependent switching in ferromagnetic Hall crosses," Appl. Phys. Lett. **108**, 092405 (2016).*
 - We indeed realize that we have not explicitly cited any works that show the use of the AHE to detect AOS, although we also note that we do not claim to have novelty on this part in our work. Nevertheless, we added a citation to the work listed under *a* accordingly.

Date

15 October 2018

Page

7 of 7

In the case of the work listed under *b*, we already cite the more well-known follow-up work by the authors in which they use the same AHE technique to measure the AOS in a Hall cross [6]. In our revised manuscript, we added a citation to this paper in relation to the AHE measurement of AOS.

[6] M.S. El Hadri et al., Phys. Rev. B **94**, 064412 (2016)

Change made to manuscript: References *a* and [6] are added to the manuscript, line 96.

Reviewers' comments:

Reviewer #2 (Remarks to the Author):

Combining AOS with Racetrack memory is an interesting attempt, however it is still not obvious whether this combination brings practice advantages or uses importantly new physics that enables their results. At this stage, I can't recommend this manuscript for publication.

Considering the slow domain wall motion (~ 7 m/s) they demonstrated, I'm not convinced by their claim that their approach provides an entirely new class of hybrid photonic-magnetic devices in which information can be directly coupled from the photonic to the magnetic domain without any intermediate electronic steps. First, replacing an electronic writing transducer with an optical system containing fs laser is not well justified as a practical advantage. Also directly writing the photonic information onto the magnetic domain can be achieved using AOS and the shifting of the magnetic domain can be achieved reliably using mechanical motions such as spinning disks in the hard disk drives where a velocity of 10s of m/s and linear density of ~ 10 nm/bit.

What prevent the authors to actuate the domain wall faster than 7 m/s? The authors simply stated that they use these low velocities to allow for a straight forward and reliable determination of the SHE efficiency and a clear interpretation of the concept, which is not logically convincing.

For the downsizing of the AOS towards nanometer scale, the authors claim it can be solved using HAMR, however, AOS and HAMR use quite different media and recording dynamics. It is not justified to build direct connections between them.

Reviewer #3 (Remarks to the Author):

I'm now more convinced that demonstrated all-optical writing in a racetrack memory is an important development in all-optical switching (AOS). With the potential of much faster domain wall velocity, it may lead to a practical technology, though the concept of integrated opto-spintronic devices that combines spintronics and photonics technologies has been demonstrated in previous works, such as:

All-Optical Switching of Magnetic Tunnel Junctions with Single Subpicosecond Laser Pulses, Jun-Yang Chen, Li He, Jian-Ping Wang, and Mo Li, Phys. Rev. Applied 7, 021001 (2017)

Nevertheless, this reviewer now supports its publication in Nature Communications.

Subject
Rebuttal

Date
13 November 2018

Response to the reviewer's comments:

Reviewer #2 (Remarks to the Author):

Combining AOS with Racetrack memory is an interesting attempt, however it is still not obvious whether this combination brings practice advantages or uses importantly new physics that enables their results. At this stage, I can't recommend this manuscript for publication.

We thank **reviewer #2** for his/her analysis of our work, and we would like to provide a point-by-point response to the reviewer's comments. Changes made to the manuscript are mentioned below, and are highlighted in the revised manuscript.

1. *Considering the slow domain wall motion (~ 7 m/s) they demonstrated, I'm not convinced by their claim that their approach provides an entirely new class of hybrid photonic-magnetic devices in which information can be directly coupled from the photonic to the magnetic domain without any intermediate electronic steps. First, replacing an electronic writing transducer with an optical system containing fs laser is not well justified as a practical advantage. Also directly writing the photonic information onto the magnetic domain can be achieved using AOS and the shifting of the magnetic domain can be achieved reliably using mechanical motions such as spinning disks in the hard disk drives where a velocity of 10s of m/s and linear density of ~ 10 nm/bit.*
- We believe that the hybrid photonic-magnetic device can be of vital importance in the field of photonic integration, where it is used as a (buffer) memory device or a shift register in photonic integrated circuits (i.e., on-chip), and not per see as a stand-alone data storage device, as suggested by the reviewer (e.g., using AOS in combination with existing hard-disk-drive technology).

In this regard, the possibility to omit the intermediate electronic steps when writing the optical data in the magnetic domain is of great value. Using electronic transducers, the optical data needs to be converted to an electrical signal, which in turn needs to be converted to the magnetic domain, all reducing both the speed and energy-efficiency of the writing process.

For completeness, we emphasize that it is not our idea to ‘integrate’ an external fs laser, but use on-chip data signals propagating through the photonic waveguides. In this context it is relevant to mention that AOS has been demonstrated for ~ 10 ps pulses, while a switching energy of 10 fJ at a data rate of 10 GHz would translate to an average power of no more than 0.1 mW.

Change made to manuscript: To highlight the vision of the use of the hybrid structure in photonic integrated circuits, the sentence in line 36 is changed to: “, showing high potential to be used in future photonic integrated circuits.”

2. *What prevent the authors to actuate the domain wall faster than 7 m/s? The authors simply stated that they use these low velocities to allow for a straight forward and reliable determination of the SHE efficiency and a clear interpretation of the concept, which is not logically convincing.*
- The statement on the use of the low velocities to allow for a straightforward and reliable interpretation of the SHE efficiency and on-the-fly AOS measurements is with regard to the low velocities of ≈ 100 $\mu\text{m/s}$ used in the main manuscript. The measurement showing a domain wall velocity of 7 m/s show that much higher velocities in the m/s range are already achievable with current densities still an order of magnitude smaller than used for the highest DW velocities up to 1000 m/s in other works, and without structural or ambient temperature optimization. There is no physical limitation that prevents these high domain wall velocities in the presented structures, however, this would be out of the scope of the presented work, and is believed to have no additional value to the demonstration of the basic concept.
3. *For the downsizing of the AOS towards nanometer scale, the authors claim it can be solved using HAMR, however, AOS and HAMR use quite different media and recording dynamics. It is not justified to build direct connections between them.*

We do not propose that the downscaling can be achieved using the complete recording mechanism used in HAMR, only the very local heating used in HAMR is of interest.

As demonstrated by the toggle-switching using linearly polarized laser pulses in the manuscript, the single-pulse AOS is a purely thermal process. Therefore, AOS can be scaled down using plasmonic antennas that can reduce the laser-pulse spot size to the nm scale. This technique is already well established in HAMR, where these plasmonic antennas are used to locally heat the magnetic material with a FWHM spot size < 40 nm. For this reason we mention the plasmonic antenna’s used in HAMR with respect to the downscaling in the manuscript, line 328: “Moreover, due to the thermal nature of the AOS, the final downsizing of the AOS towards the nanometer scale can be done using plasmonic antenna's, a technique already used in heat assisted magnetic recording (HAMR) to very locally

Date
13 November 2018

heat the recording material by reducing the FWHM laser-pulse size to < 40 nm.”

Page
3 of 3

Reviewer #3 (Remarks to the Author):

I'm now more convinced that demonstrated all-optical writing in a racetrack memory is an important development in all-optical switching (AOS). With the potential of much faster domain wall velocity, it may lead to a practical technology, though the concept of integrated opto-spintronic devices that combines spintronics and photonics technologies has been demonstrated in previous works, such as: All-Optical Switching of Magnetic Tunnel Junctions with Single Subpicosecond Laser Pulses, Jun-Yang Chen, Li He, Jian-Ping Wang, and Mo Li, Phys. Rev. Applied 7, 021001 (2017)

Nevertheless, this reviewer now supports its publication in Nature Communications.

We thank **reviewer #3** for his/her analysis of our work, and are pleased to hear that he/she supports the publication of the manuscript in Nature Communications.

REVIEWERS' COMMENTS:

Reviewer #2 (Remarks to the Author):

At this point, we don't have comments about the technical contents. We are not fully convinced about the importance of combining AOS with racetrack recording for its possibility in the application in future optical circuits, however the technique itself might be technically interesting for researchers in the field of magnetic recording. We are OK to accept this manuscript.